# Expression of miR-24-1-5p in Tumor Tissue Influences Prostate Cancer Recurrence: The PROCA-*life* Study

**DOI:** 10.3390/cancers14051142

**Published:** 2022-02-23

**Authors:** Einar Stikbakke, Tom Wilsgaard, Hege Sagstuen Haugnes, Mona Irene Pedersen, Tore Knutsen, Martin Støyten, Edward Giovannucci, Anne Elise Eggen, Inger Thune, Elin Richardsen

**Affiliations:** 1Department of Clinical Medicine, Faculty of Health Sciences, UiT, The Arctic University of Norway, 9037 Tromsø, Norway; hege.sagstuen.haugnes@unn.no (H.S.H.); tore.knutsen@unn.no (T.K.); martin.stoyten@uit.no (M.S.); inger.thune@uit.no (I.T.); 2Department of Oncology, University Hospital of North Norway, 9038 Tromsø, Norway; 3Department of Community Medicine, Faculty of Health Sciences, UiT, The Arctic University of Norway, 9037 Tromsø, Norway; tom.wilsgaard@uit.no (T.W.); anne.elise.eggen@uit.no (A.E.E.); 4Translational Cancer Research Group, Institute of Clinical Medicine, UiT, The Arctic University of Norway, 9037 Tromsø, Norway; mona.i.pedersen@uit.no (M.I.P.); elin.richardsen@unn.no (E.R.); 5Department of Urology, University Hospital of North Norway, 9038 Tromsø, Norway; 6Department of Medicine, Brigham and Women’s Hospital, Harvard Medical School, Boston, MA 02115, USA; egiovann@hsph.harvard.edu; 7Departments of Nutrition and Epidemiology, Harvard T.H. Chan School of Public Health, Boston, MA 02115, USA; 8Institute of Clinical Medicine, Faculty of Medicine, University of Oslo, 0316 Oslo, Norway; 9Department of Oncology, The Cancer Centre, Oslo University Hospital, 0424 Oslo, Norway; 10Department of Medical Biology, Faculty of Health Sciences, UiT, The Arctic University of Norway, 9037 Tromsø, Norway; 11Department of Clinical Pathology, University Hospital of North Norway, 9038 Tromsø, Norway

**Keywords:** prostate cancer, microRNA, miR-24-1-5p, radical prostatectomy, biomarker, population study

## Abstract

**Simple Summary:**

Prostate cancer is a major cause of health loss and death worldwide, and better tools to assess risk levels in individual patients are needed. MicroRNAs (miRNAs) are small molecules with critical regulatory roles in cell functions and are also involved in prostate cancer development. The aim for this study was to investigate the role of miR-24-1-5p regarding prognosis in men diagnosed with prostate cancer and treated with radical prostatectomy. We collected prostate cancer tissue from 142 men already enrolled in a population-based cohort study who underwent prostatectomy. We examined the tissue expression of miR-24-1-5p in prostate cancer using in situ hybridization (ISH) and semi-quantitative scoring. We found that a high miR-24-1-5p expression was associated with a doubled risk of recurrence of prostate cancer.

**Abstract:**

The role of miR-24-1-5p and its prognostic implications associated with prostate cancer are mainly unknown. In a population-based cohort, the Prostate Cancer Study throughout life (PROCA-*life*), all men had a general health examination at study entry and were followed between 1994 and 2016. Patients with available tissue samples after a prostatectomy with curative intent were identified (*n* = 189). The tissue expression of miR-24-1-5p in prostate cancer was examined by in situ hybridization (ISH) in tissue microarray (TMA) blocks by semi-quantitative scoring by two independent investigators. Multivariable Cox regression models were used to study the associations between miR-24-1-5p expression and prostate cancer recurrence. The prostate cancer patients had a median age of 65.0 years (range 47–75 years). The Cancer of the Prostate Risk Assessment Postsurgical Score, International Society of Urological Pathology grade group, and European Association of Urology Risk group were all significant prognostic factors for five-year recurrence-free survival (*p* < 0.001). Prostate cancer patients with a high miR-24-1-5p expression (≥1.57) in the tissue had a doubled risk of recurrence compared to patients with low expression (HR 1.99, 95% CI 1.13–3.51). Our study suggests that a high expression of miR-24-1-5p is associated with an increased risk of recurrence of prostate cancer after radical prostatectomy, which points to the potential diagnostic and therapeutic value of detecting miR-24-1-5p in prostate cancer cases.

## 1. Introduction

Prostate cancer (PCa) is a major cause of health loss and death worldwide, and it is a heterogeneous disease [1,2]. Compared with localized low-risk PCa that can be actively surveyed without management, the treatment for aggressive high-risk PCa is most often systemic and complex. We need valid prognostic biomarkers to distinguish low-risk indolent PCa from aggressive PCa.

MicroRNAs (miRNAs) are a class of endogenous non-coding small RNA molecules associated with the regulation of gene expression and are “fine-tuners” of the immune system [3]. These have been studied for their potential to serve as molecular prognostic biomarkers for cancer including PCa [4]. In particular, differential miRNAs’ expression profiles between tumour and normal tissues have been observed for PCa, as well as for other other cancer types [4,5]. In a recent systematic review, fifteen miRNAs were associated with PCa prognosis [4]. These are transcribed as ~70 nucleotide precursors in a stem-loop sequence and are subsequently processed by the Dicer enzyme to give two mature ~22 nucleotide products. These miRNAs bind to the 3^/^-untranslated region (3^/^-UTR) of target messenger RNA (mRNA) and are used to identify target mRNA transcripts. They can prevent protein expression through cleavage of specific target mRNAs or through inhibition of their translation, and thus influence developmental processes, tissue housekeeping and tumorigenesis [6]. Aberrant expression or dysregulation of miRNA can influence the activity of tumor suppressors or oncogenes in many human cancers [6,7], including prostate cancer [8]. An example of this is how miR-21 expression can trigger an epithelial to mesenchymal transition in aggressive prostate cancer cells through the Wnt signaling axis [9].

Additionally, miRNAs have also been associated with the tumor microenvironment, as well as PD-L1 and STAT3 signaling in prostate cancer cells, supporting the idea that miRNAs play a role in and are linked to inflammation [10]. Most prostate tumors contain immune cells, and chronic inflammation, one of the hallmarks of cancer development [11,12], has been proposed as a key factor in prostate cancer development [13,14,15]. The suggested hypothesis is partly based on observations of inflammatory cells in the prostate microenvironment of adult men and partly by the observation that this inflammation has been associated with precursor lesions in the prostate gland, termed proliferative inflammatory atrophy [16,17,18,19]. However, much remains unknown regarding possible biological mechanisms operating in relation to prostate cancer development and systemic and local inflammation, and only several mechanisms, including miRNAs and factors related to the immune system, have been studied [3,20].

The effects of miRNAs in prostate cancer have been studied, but the biological mechanisms operating, as well as the types of miRNAs and their functions, have not yet been clarified [4,6,21,22]. Importantly, no prostate-specific miRNAs have yet been definitively identified. We previously studied the association between several miRNAs and prostate cancer recurrence and survival [23,24,25,26,27,28]. High expressions of miR-205, miR-17-5p, miR-20a-5p, miR-210, and miR-141 and a low expression of miR-424 were all associated with an increased risk of prostate cancer recurrence. These miRNAs have been suggested to be associated with inflammation; however, there is limited knowledge [3]. Furthermore, few have investigated the association between miR-24 and prostate cancer [8]. Through deep sequencing of prostatectomy specimens, it was observed that miR-24 was downregulated compared to non-cancer prostate tissue [29]. Another study, by Hashimoto et al. found that miR-24 was differentially expressed in African American and Caucasian American prostate cancer patients [30]. Interestingly, miR-24-3p enhanced Paclitaxel sensitivity in Paclitaxel-resistant prostate cancer cells [31], while in xenograft cell lines, miR-24 was downregulated in metastatic prostate cancer compared to non-metastatic [32]. Furthermore, the miR-24 expression was significantly lower in prostate cancer cell lines compared to a normal prostate epithelial cell line. These findings suggest that miR-24 plays a tumor suppressor role in prostate cancer and targets p27 and p16 in prostate cancer cells [33]. Current knowledge about miR-24 is largely based on in vitro studies and/or mouse models. The stem-loop sequence hsa-miR-24-1 is the processor of two mature sequences: hsa-miR-24-1-5p and hsa-miR-24-3p [34]. To our knowledge, previous studies have not reported which sequences of miR-24 they have used [32,33].

The present study is based on men participating in the Tromsø Study, a population-based cohort study, which has a high attendance proportion and long follow-up time [35]. Complete information on prostate cancer cases, including detailed medical and pathological records, has been obtained in a substudy, the Prostate Cancer Study throughout life (PROCA-*life*) [36]. The role of miR-24s, including the different types of miR-24 and their prognostic implications, is still under debate, and their potential diagnostic and therapeutic values are not clarified. Therefore, the main aim of the present study was to analyze the influence of miR-24-1-5p regarding aggressiveness and prognosis in men diagnosed with prostate cancer and treated with radical prostatectomy.

## 2. Materials and Methods

### 2.1. Study Sample

The present study cohort, PROCA-*life* study, is based solely on men aged ≥ 25 years who were enrolled in the population-based Tromsø Study from 1994 to 2016 (Tromsø 4, 1994–95, Tromsø 5, 2001, Tromsø 6, 2007–2008, Tromsø 7, 2015–2016) [37]. The procedures for invitations, screening, and examinations were almost identical in all three surveys. Moreover, all data collection was performed by trained research technicians at one study site. Age-eligible men were invited to participate by a personal invitation [35,37]. A total 75.6% of invited men attended, completed questionnaires, and provided biological specimen samples and clinical measurements.

### 2.2. Questionnaires, Clinical Assessments, and Assessment of Lipids and PSA

Height and weight were measured on an electronic scale with the participants wearing light clothing and no shoes. Height was measured to the nearest 1 cm (cm) in Tromsø 4 and nearest 0.1 cm in Tromsø 5–7. Weight was measured to the nearest 500 g in Tromsø 4 and nearest 100 g in Tromsø 5–7. Body mass Index (BMI) was calculated using the formula weight/height^2^ (kg/m^2^). Blood pressure (BP) was measured on the right arm three times at one-minute intervals after two minutes of seated rest, and the mean of the last two measurements was used. Information about lifestyle factors was obtained from the questionnaires. Alcohol consumption was defined as more than 1 unit (drink) of alcohol per month, as described by others in the same cohort [38,39].

Blood samples were drawn by trained research assistants on attendance at each survey and were non-fasting. Analyses of serum samples were completed at the Department of Laboratory Medicine, University Hospital of Northern Norway (UNN), Tromsø, Norway [35]. For white blood cell count (WBC), 5 mL of blood was collected into Vacutainer tubes containing K3-EDTA 40 lL, 0.37 mol/L per tube, and analyzed within 12 h by an automated blood cell counter (Coulter CounterÒ and Coulter LH750 Coulter Electronics, Luton, UK). Total cholesterol and triglyceride levels were analyzed by enzymatic colorimetric methods with commercially available kits (CHOD-PAP for cholesterol). Prostate Specific Antigen (PSA) measurements were taken for prostate cancer cases only, as a part of the clinical routine for diagnosis and follow-up (1990–1994 Stratus^®^ PSA Fluorometric Enzyme Immunoassay, (BDI, Miami, FL, USA), 1994–2001 AxSYM Psa Reagent Pack, (Abbott^®^, Lake Bluff, IL, USA), 2001–2020 Bayer^®^ PSA Reagent Pack Immuno I (Prod. Nr. T01-3450-51, Technicon Immuno I (New York, NY, USA).

### 2.3. Identification of Prostate Cancer Cases and Detailed Medical History during Follow-Up

Prostate cancer cases during follow-up (until 31 December 2018) were identified in the Cancer Registry of Norway (*n* = 947), by using the unique national 11-digit identification numbers (The National Population Registry at Statistics Norway). Cases with available tissue samples after prostatectomy with curative intent were identified by cross-linkage with the archive of Department of Clinical Pathology, University Hospital of North Norway, Tromsø, Norway (*n* = 189), and these constituted the eligible study population in the current study (see flow chart figure in Appendix B). Overall, 43 cases were not technically successful in the in situ hybridization (ISH) staining process and were excluded. Furthermore, four cases were excluded because they did not have curative surgery, leaving a final study population of 142 men (Figure A1).

Detailed clinical information was obtained by trained physicians (MS, TK, and ES) and included prostate cancer treatments and recurrence. Cause of death was obtained through linkage with the Norwegian Death Registry by use of the unique personal identification number. Most of the prostate cancer patients (88.7%) underwent prostatectomy a few months after being diagnosed; the remainder of the study population (11.2%) underwent active surveillance until their prostate cancer showed signs of increasing aggressiveness. Date of prostatectomy was used for calculation of age and follow-up time. The current study is based on the Tromsø Study survey closest to the date of prostatectomy for baseline data such as height, weight, blood pressure, triglyceride levels, and alcohol use.

Histopathological information was obtained from medical records, but all histopathological specimens were re-examined by one specialized uropathologist (ER) and classified according to the latest International Society of Urological Pathology (ISUP) guidelines using their Gleason scores and ISUP grade groups [40]. Prostate cancer cases were divided into three risk groups based on PSA level at diagnosis, highest ISUP grade group, and clinical T-stage, according to classification guidelines from the European Association of Urology (EAU) [41]. Risk group 1 (low) was defined as: PSA <10 µg/L, clinical T-stage (cT-) 1, and ISUP grade group 1. Risk group 2 (intermediate) was defined as: PSA: 10–20 µg/L, cT-stage 2, or ISUP grade group 2–3. Risk group 3 (high) was defined as: PSA: >20–100 µg/L, cT-stage 3, or ISUP grade group 4–5. ISUP grade groups were reported after reclassification when available. Cancer of the Prostate Risk Assessment Postsurgical Score (CAPRA-S Score), a validated score developed to predict outcomes after radical prostatectomy, was also used to classify patients into risk groups [42]. This score is based on surgical margin, seminal vesicle invasion, extracapsular extension, lymph node invasion, PSA value, and Gleason/ISUP Grade Groups.

### 2.4. Microarray Construction

Tissue microarrays (TMAs) were constructed for the analysis of ISH staining expression. For each case, one uropathologist (ER) identified and marked representative areas of the prostate specimens with tumor epithelial cells (TE) and normal epithelial cells (NE). From each of these areas, 0.6 mm cores were sampled from each donor block and inserted into paraffin blocks to construct TMA blocks by using a tissue-arraying instrument (Beecher Instruments, Silver Springs, MD, USA). The details of the technique were described earlier [43].

### 2.5. In Situ Hybridization (ISH)

The tissue expression of mature miR-24-1-5p in prostate cancer was examined by in situ hybridization (ISH). The principle of the method is based on the ability of specific microRNA locked nucleic acid (LNA) probes to bind to target microRNA in tissue followed by chromogenic visualization. ISH staining was performed automatically in a Ventana Discovery Ultra instrument. Necessary efforts to avoid RNA degradation in tissue were made in work routines and by using RNAse-free buffers during the process.

### 2.6. Optimization and Validation

LNA probe concentrations, hybridization temperatures, and incubation times were optimized before staining the tissue of interest. Target retrieval treatment was adjusted to improve availability of microRNA sequence for the target and control probes. A TMA multi organ block with several normal and tumor tissues was used for optimization of the ISH method and validation of miR-24-1 expression in different tissues. We used a U6snRNA probe as a positive control and to ensure the sensitivity level of the method. Strong nuclear U6snRNA staining also indicates a low degree of RNA degradation of the tissue. A scramble miRNA negative control probe showed no unspecific staining. Optimized ISH parameters are presented in Table A1, and details of the products are ordered in Table A2.

External validation of the LNA probes was completed by supplier company QIAGEN. The LNA miRNA probes were purified by HPLC (High-Performance Liquid Chromatography) and analyzed by Capillary Electrophoresis or HPLC. The identity of compounds was confirmed by using Mass Spectrometry. For more details on the ISH procedure, see Appendix A.

### 2.7. Scoring

The expression of miR-24-1-5p was assessed by semi-quantitative scoring by two trained independent investigators (ES, ER). The color intensity was graded as negative (0), weak (1), moderate (2), strong (3), or missing (4) (Figure 1). Two areas of TE cells and two areas of NE cells were scored for each patient. The same methodology has been used by our group previously [24,28]. Stromal areas were not scored due to little positivity. Mean and median scores were calculated for TE and for NE separately, as well as for TE+NE combined. High expression of miR-24-1-5p was defined as a score equal to or higher than the median score of the study population. Inter-observer variability was assessed by calculating linearly weighted Kappa statistics and showed a moderate agreement (Kappa 0.59 (SD 0.50–0.68)).

The primary endpoint was defined as a composite endpoint, including any evidence of recurrent prostate cancer after surgery, biochemical failure (PSA-level ≥0.2), and/or clinical/radiological signs of prostate cancer defined by the treating physician. Endpoints were updated until August 2021.

### 2.8. Statistical Methods

Selected characteristics that describe the study population are presented as a mean (standard deviation), median (range) or percent (numbers). Spearman’s correlation coefficient was used for correlation analysis between miR-24-1-5p and clinicopathological markers. The five-year recurrence-free percentage was calculated using the Kaplan–Meier survival function, and statistical differences between different groups (e.g., ISUP grade group, EAU risk group, CAPRA-S) were tested by using log-rank test.

Multivariable Cox proportional hazard models, with time after surgery as timescale, were used to study whether miR-24-1-5p and clinicopathological markers were independently associated with a risk of prostate cancer recurrence. Several variables were assessed as potential confounders based on suggested biological mechanisms and/or significant associations in unadjusted models. Age at surgery (continuous), CAPRA-S (categorical), BMI (continuous), alcohol habits (categorical), and cholesterol levels (continuous) were included in the final models as covariates. We performed a stratified analysis via systolic blood pressure based on previous observations suggesting that elevated systolic blood pressure is associated with prostate cancer risk [44]. The proportional hazard assumption was assessed by visually controlling that the log minus log survival curves were parallel. The Kaplan-Meyer method was used for drawing survival plots for high vs. low expression of miR-24-1-5p. We conducted all statistical tests with STATA/MP version 16 (StataCorp LLC, College station, TX, USA) and used a two-sided significance level of *p* < 0.05.

## 3. Results

### 3.1. Patient Characteristics

The 142 men that constituted the study population entered the Tromsø Study on average 8.0 years before prostatectomy. The median age at prostate cancer diagnosis was 64 years (range 46–74 years), the median age at prostatectomy was 65 years (47–75 years), and prostatectomy was performed between 2001 and 2018 (Table 1). The prostate cancer patients had an average BMI of 27.1 kg/m^2^, systolic BP of 134.9 mmHg (SD 16.8), and diastolic BP of 80.4 mmHg (SD 9.4) at study entry. A total of 61.3% of the prostate cancer patients had a systolic blood pressure higher than 130 mmHg. The mean level of white blood cells was 6.60 × 10^9^/L (SD 1.67), total cholesterol was 5.78 mmol/L (SD 1.12), and triglyceride level was 1.70 mmol/L (SD 0.90). Additionally, 46.1% were alcohol users.

The surgical technique changed during the study period: 47.2% of the patients had open (retropubic or perineal) prostatectomy, mostly before the year 2012, while 52.8% had laparoscopic prostatectomy (manual or robot-assisted). Lymph node dissection was performed in 36.6% of the patients. The mean PSA at prostate cancer diagnosis was 10.5 ng/mL (SD 9.5). The histopathologic tumor stage was pT2c for 47.9% of the patients, while 26.1% had pT3, and the ISUP grade group was 1 or 2 for 73.8% of the patients. The median CAPRA-S score was 3 (39.4% Capra-S low (0–2), 46.5% Capra-S intermediate (3–5), and 14.1% Capra-S high (6–12). Positive surgical margins were found in 30.5% of the cases. Overall, 26.9% of the prostate cancer patients had a relapse after prostatectomy during follow-up (until August 2021).

### 3.2. miR-24-1-5p Expression

The mean score for miR-24-1-5p expression was 1.60 in TE cells, 1.35 in NE cells, and 1.49 in TE and NE cells combined (Table 2). The median value was used as cut-off value for high miR-24-1-5p expression and was ≥1.67 in TE and ≥1.50 in NE. The cut-off value for high TE+NE combined was ≥1.57. In the total population, 43.7% had high TE+NE, 43.7% had high TE, and 45.1% had high NE.

### 3.3. miR-24-1-5p Correlations

The level of white blood cells at study entry (pre-diagnostic) correlated with miR-24-1-5p expression in both TE and NE (r = 0.21, *p* = 0.02 and r = −0.21, *p* = 0.01, respectively). Furthermore, BMI and triglyceride levels at study entry correlated with miR-24-1-5p expression in NE (r = −0.27, *p* = 0.01 and r = −0.24, *p* = 0.006). Positive surgical margin correlated with miR-24-1-5p expression in TE (r = 0.19, *p* = 0.029). CAPRA-S correlated with miR-24-1-5p expression in TE (r = 0.21, p = 0.020) (results not presented in table). There were no correlations between miR-24-1-5p expression and PSA at diagnosis, Gleason score, perineural invasion, age at surgery, or BMI.

### 3.4. Recurrence-Free Survival

Age at surgery was not associated with recurrence-free survival (Table 3). Increasing CAPRA-S score, ISUP grade group, and EAU risk group were all significant prognostic factors for decreasing five-year recurrence-free survival (*p* < 0.001). Our data suggested a higher number of recurrences in the group with high expression of miR-24-1-5p but of borderline significance (*p* = 0.098) (Figure 2). In the subgroup of prostate cancer patients with high pre-diagnostic systolic blood pressure (≥130 mmHg), high expression of miR-24-1-5p was a prognostic factor for recurrence.

### 3.5. Multivariable Analyses

In our multivariable model, we adjusted for age, Capra-S group, BMI, cholesterol level, and alcohol use, based on suggested biological mechanisms. High miR-24 expression in the tissue (TE + NE) was associated with an almost doubled risk of the recurrence of prostate cancer compared to that with low miR-24-1-5p expression (HR 1.99, 95% CI 1.13–3.51) (Table 4). The results were also observed in the subgroup of prostate cancer patients with high pre-diagnostic systolic blood pressure. There was no significant interaction between miR-24 expression and blood pressure, nor between miR-24 expression and follow-up time.

## 4. Discussion

We found that high expression of miR-24-1-5p was associated with an almost doubled risk of recurrence (biochemical or clinical) after radical prostatectomy, when adjusting for known histopathological risk factors. We were also able to adjust for known lifestyle risk factors due to the pre-diagnostic information assessed at the study entry. We did not observe correlations between age, perineural infiltration, PSA values, or Gleason score and miR-24-1-5p expression, however it did correlate with CAPRA-S, a score that incorporates the PSA and Gleason score. Of note, we observed positive surgical margins correlated with miR-24-1-5p expression.

Prostate cancer is a heterogeneous condition, ranging from indolent to life-threatening, and we need better tools for disease stratification. Development of biomarkers for risk stratification, personalized treatment, and follow-up is needed. Other miRNAs have shown good correlation between levels in tissue and in blood or urine, and the development of liquid biomarkers would be a great advantage for the patient by limiting the need for invasive tissue biopsies. In addition, miRNAs possibly play a role in prostate cancer metastasis and are therefore potential targets for new therapeutic agents [45].

To our knowledge, this is the first study to investigate whether the expression of miR-24-1-5p in prostate cancer tissue is associated with prognosis. Our findings are in part supported by others, although few studies have investigated the role of miR-24-1-5p in prostate cancer. Most of these studies have been experimental. A recent meta-analysis studied the prognostic significance of miR-24 in various cancers and found that high miR-24 expression was associated with poor overall survival [46]. The meta-analysis consisted of 17 studies, and a total of 1705 patients, of whom none had prostate cancer. Another recent study observed that the expression of miR-24-1-5p decreased 16-fold after radiotherapy doses of 6 and 7 Gy in prostate cancer cell lines treated with radiation, suggesting that expression of miR-24-1-5p may impact the efficacy of important treatment modalities, such as radiation therapy [47]. Further studies are needed to explore the importance of this observation.

A few studies have evaluated the other mature sequence of miR-24, miR-24-3p, which has been suggested as a diagnostic biomarker for prostate cancer in serum [48,49]. The circRNA protein kinase C-iota has been suggested to influence tumor development, and a study found this molecule triggers growth and metastasis in prostate cancer by downregulation of miR-24-3p [50]. ACVR1B, BCL2, BIM, eNOS, FGFR3, JPH2, MEN1, MYC, p16, and ST7L are miR-24 targets that have been experimentally validated in human cells [20]. However, it is unclear whether these results will be valid for the association between miR-24-1-5p and prostate cancer development.

The relationship between prostate cancer and inflammation has been the subject of several studies. Inflammation is one of the classic hallmarks of cancer [11], and inflammatory cells associated with the precursor lesions for prostate cancer in the prostate gland have been observed [13]. We have previously discovered that systemic pre-diagnostic inflammatory biomarkers were associated with prostate cancer development [36]. In mouse models, prostate-specific PTen deletion has been found to activate inflammatory microRNA expression pathways [51]. Additionally, miR-24 has been linked to inflammation [3]: miR-24 was found to regulate phagocytosis in myeloid inflammatory cells [52]. In a murine model, miR-24 was a central regulator of vascular inflammation [53]. In a model with primary human macrophages, miR-24 would produce anti-inflammatory action by inhibiting the production of pro-inflammatory cytokines, and these results suggest that overexpression of miR-24 would have mostly anti-inflammatory effects [54]. Conversely, miR-24 belongs to the miR-23~27~24 cluster, and this cluster has been shown to reduce TNF-α and IL-6 production [55]. In summary, the current literature on miR-24 is not consistent on whether miR-24 has a pro- or anti-inflammatory role. Our observation that the association between miR-24-1-5p and prostate cancer recurrence was suggestively more pronounced among the prostate cancer patients with high pre-diagnostic systolic blood pressure supports the possibility of a role associated with low-grade systemic inflammation.

The strengths of our study include the broad pre-diagnostic information about the participating prostate cancer patients, a relatively large sample of patients with prostate cancer prostatectomy specimen (*n* = 142), as well as detailed histopathological and medical records for all the patients. The methodology for TMA-production and in situ hybridization has been used in our lab for several tissues and is well tested [23,24,25,56,57]. Scoring of miR-24-1-5p was completed by two independent observers and showed a moderate inter-observer variability. Earlier studies have focused on murine models and cell lines, while our study uses human prostate cancer tissue, which is in line with future clinical studies. Our study also had some weaknesses. The sample size was not large enough for subgroup analysis, and 42 samples were lost due to technical problems in the ISH-process. The scoring of miRNA-expression was semi-quantitative and thus subject to variability. We only had prostate tissue available and were not able to test the expression of miR-24-1-5p in other samples such as serum or urine.

## 5. Conclusions

Our study suggests that high expression of miR-24-1-5p is associated with an increased risk of failure after radical prostatectomy, as well as when adjusting for known histopathological risk factors. The results are experimental, based on a relatively small sample size, and should be interpreted with caution. Nevertheless, this could be a steppingstone to further research about the role of miR-24 in prostate cancer and possibly a future tool for better risk stratification.

## Figures and Tables

**Figure 1 cancers-14-01142-f001:**
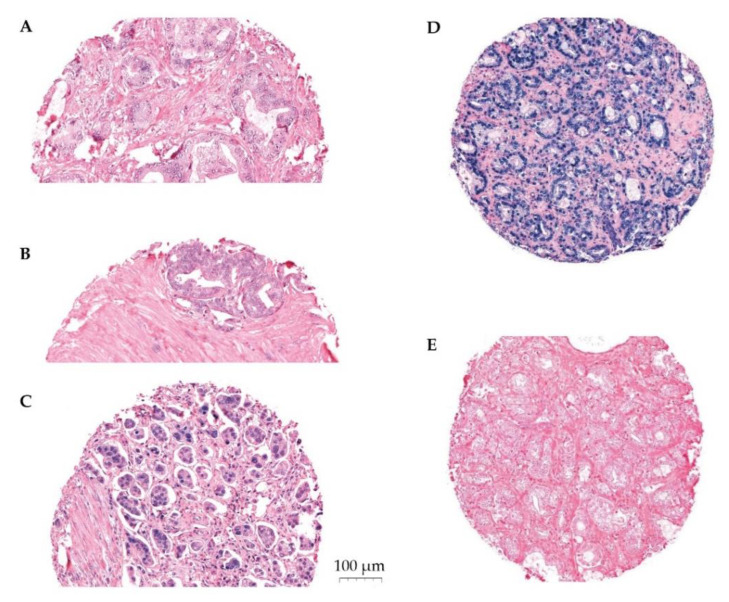
Panel of ISH stained cores. Representative scoring of miR-24-1-5p in tumor epithelium (TE): (**A**) weak expression, (**B**) moderate expression, (**C**) strong expression, (**D**) U6 positive control staining, and (**E**) scrambled miRNA negative control staining. The PROCA-life study.

**Figure 2 cancers-14-01142-f002:**
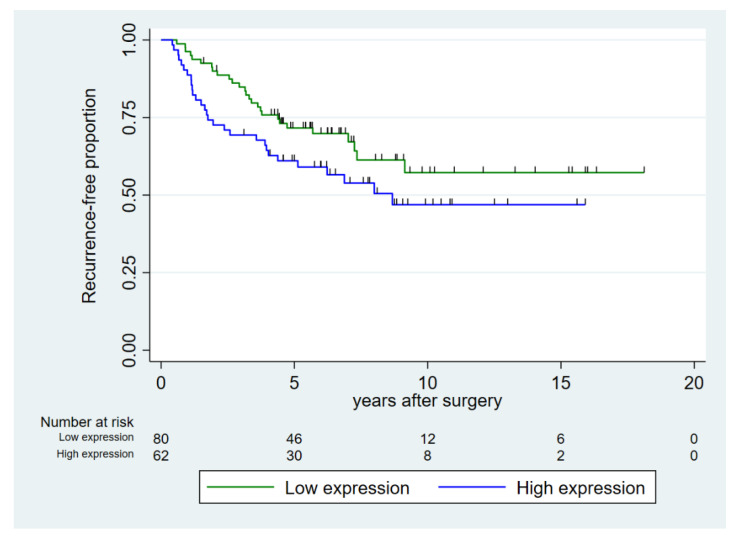
Recurrence-free proportion of prostate cancer after prostatectomy, dichotomized into high vs. low expression of miR-24-1-5p in prostate cancer tissue (tumor epithelium and normal epithelium combined (TE+NE)). Low expression was defined as score < 1.57 and high expression as score ≥ 1.57.

**Table 1 cancers-14-01142-t001:** Distribution of selected characteristics among the prostate cancer patients who received prostatectomy in the PROCA-life Study (1994–2018).

Characteristics	Prostatectomy Cases (*n* = 142)
Age at Study Entry, Median, Range (Years)	58.5 (34–73)
Birth Year Median, Range (Year)	1947 (1934–1967)
Age at Surgery, Median, Range (Years)	65.0 (47–75)
Observation Time from Study Entry to Surgery (Years)	8.0 (6.6)
Observation Time from Surgery to End of Follow-Up (Years)	4.8 (3.4)
PSA at Diagnosis (μg/L)	10.5 (9.5)
Relapse Rate (Biochemical + Clinical), % (n)	26.9 (38)
*Clinical Assessments at Study Entry*	
Body Mass Index (kg/m^2^)	27.1 (3.15)
Systolic Blood Pressure (mmHg)	134.9 (16.8)
Diastolic Blood Pressure (mmHg)	80.4 (9.4)
White Blood Cells (×10^9^/L)	6.60 (1.67)
Total Cholesterol (mmol/L)	5.78 (1.12)
Triglyceride (mmol/L)	1.70 (0.90)
Alcohol Intake (>1 Unit of Alcohol per Month), % *(n)*	46.1 (65)
*Surgical Technique, % (n)*	
Open Prostatectomy, Retropubic	38.0 (54)
Open Prostatectomy, Perineal	9.2 (13)
Laparoscopic Prostatectomy	6.3 (9)
Robotic-Assisted Laparoscopic Prostatectomy (RALP)	46.5 (66)
Lymph Node Dissection Performed, % (n)	36.6 (52)
*Histopathological Stage, % (n)*	
pT2a	17.0 (24)
pT2b	8.5 (12)
pT2c	48.2 (68)
pT3a	16.3 (23)
pT3b	9.9 (14)
*ISUP Grade Group, % (n)*	
1 (Gleason 3 + 3)	29.1 (41)
2 (Gleason 3 + 4)	44.7 (63)
3 (Gleason 4 + 3)	18.4 (26)
4 (Gleason 4 + 4)	6.4 (9)
5 (Gleason 4 + 5/5 + 4/5 + 5)	1.4 (2)
*Risk Group, % (n)*	
Low	25.5% (36)
Intermediate	56.0% (79)
High	18.4% (26)
*Other Histopathological Characteristics, % (n)*	
Positive Lymph Nodes (N+)	3.6 *
Ural Infiltration	21.3 (30)
Extraprostatic Growth	22.7 (32)
Normal Tissue in Surgical Margin	15.6 (22)
Positive Surgical Margin	30.5 (43)

* (5 of 52 patients with lymph node dissection). Numbers may vary due to missing information. Values are mean (standard deviation) unless otherwise specified. Abbreviations: PSA, prostate-specific antigen; ISUP, International Society of Urological Pathology. Prostate cancer risk group definitions: low: PSA < 10 µg/L, clinical T-stage (cT-) 1, and ISUP grade group 1. Intermediate: PSA: 10–20 µg/L, cT-stage 2, or ISUP grade group 2–3. High: PSA: > 20–100 µg/L, cT-stage 3, or ISUP grade group 4–5.

**Table 2 cancers-14-01142-t002:** Distribution and mean score (SD) of miR-24-1-5p expression in prostate cancer tissue by selected characteristics and their subgroups. The PROCA-life study (1994–2018).

Group	*n*	Tumor Epithelium (TE)	Normal Epithelium (NE)	Tumor + Normal Epithelium (TE+NE)
All Cases	142			
Mean Score miR-24-1-5p (SD)		1.60 (0.73)	1.35 (0.68)	1.49 (0.53)
*Distribution*				
0–0.49 Negative % (n)		3.5 (5)	7.0 (10)	2.8 (4)
0.5–1.49 Weak % (n)		30.3 (43)	40.1 (57)	39.4 (56)
1.5–2.49 Moderate % (n)		43.0 (61)	38.0 (54)	53.5 (76)
2.5–3 Strong % (n)		13.4 (19)	7.0 (10	4.2 (6)
Missing % (n)		9.9 (14)	7.8 (11)	-
*Age at Surgery*				
<65 Year	69	1.63 (0.67)	1.45 (0.64)	1.59 (0.50)
≥65 Year	73	1.58 (0.79)	1.25 (0.71)	1.40 (0.55)
*Capra-S*				
Low (0–2)	56	1.46 (0.69)	1.42 (0.76)	1.44 (0.53)
Intermediate (3–5)	66	1.69 (0.75)	1.35 (0.61)	1.57 (0.49)
High (6–12)	20	1.75 (0.73)	1.11 (0.67)	1.40 (0.67)
*Systolic Blood Pressure*				
<130 mmHg	55	1.79 (0.63)	1.38 (0.60)	1.62 (0.49)
≥130 mmHg	77	1.48 (0.77)	1.33 (0.73)	1.41 (0.55)

Numbers may vary due to missing information. Values are mean (standard deviation) unless otherwise specified. Abbreviations: CAPRA-S, Cancer of the Prostate Risk Assessment Postsurgical Score.

**Table 3 cancers-14-01142-t003:** Five-year recurrence free survival (%) for prostate cancer patients after prostatectomy by selected characteristics for all cases and by a subgroup with systolic BP ≥ 130 mmHg. The PROCA-life study (1994–2018).

Characteristics	All Cases			Cases with Pre-Diagnostic Systolic BP ≥ 130 mmHg		
	*n*	Five-Year Recurrence-Free Survival, % (95% C.I.)	*p* *	*n*	Five-Year Recurrence-Free Survival, % (95% C.I.)	*p* *
*Age at Surgery*			*0.59*			*0.27*
<65 Year	69	69.3 (56.9–78.8)		37	72.8 (55.4–84.4)	
≥65 Year	73	65.0 (52.7–74.8)		50	64.9 (49.7–76.5)	
*ISUP Grade Group*			*<0.001*			*<0.001*
1 (Gleason 3 + 3)	33	81.4 (63.1–91.2)		15	85.6 (53.3–96.2)	
2 (Gleason 3 + 4)	66	77.2 (65.1–85.6)		44	77.3 (61.9–87.1)	
3 (Gleason 4 + 3)	28	41.3 (22.8–59.0)		21	47.1 (25.1–66.4)	
4 (Gleason 4 + 4)	9	50.8 (15.7–78.1)		4	66.7 (5.4–94.5)	
5 (Gleason 4 + 5/5 + 4/5 + 5)	6	16.7 (0.8–51.7)		3	N.a	
*Risk Group*			*<0.001*			*0.0003*
Low	36	85.7 (68.9–93.8)		18	88.2 (60.2–96.9)	
Intermediate	80	70.8 (59.3–79.5)		54	71.8 (57.7–82.0)	
High	26	30.8 (14.6–48.6)		15	33.3 (12.1–56.4)	
*Capra-S*			*<0.001*			*<0.001*
Low (0–2)	56	89.2 (77.6–95.0)		33	93.9 (77.9–98.4)	
Intermediate (3–5)	66	61.2 (48.1–71.2)		41	62.6 (45.8–75.5)	
High (6–12)	20	25.0 (9.1–44.9)		13	23.1 (5.6–47.5)	
*miR-24-1-5p*			*0.098*			*0.026*
TE+NE Low	80	71.6 (60.1–80.3)		55	75.5 (61.5–85.0)	
TE+NE High	62	61.1 (47.7–72.0)		32	55.8 (37.0–71.0)	

* Log rank test for difference between groups during follow-up until study end. Numbers may vary due to missing information. Regarding miR-24-1-5p: low score was defined as <1.57 and high score ≥ 1.57. Prostate cancer risk group definitions: low: PSA < 10 µg/L, clinical T-stage (cT-) 1, and ISUP grade group 1; intermediate: PSA: 10–20 µg/L, cT-stage 2, or ISUP grade group 2–3; and high: PSA: > 20–100 µg/L, cT-stage 3, or ISUP grade group 4–5. Abbreviations: BP, blood pressure; CAPRA-S, Cancer of the Prostate Risk Assessment Postsurgical Score; ISUP, International Society of Urological Pathology; CI, Confidence Interval; TE, tumor epithelium; NE, normal epithelium.

**Table 4 cancers-14-01142-t004:** Multivariable adjusted * hazard ratio of recurrence of prostate cancer after radical prostatectomy for all cases and for a subgroup of systolic BP ≥130 mmHg. The PROCA-life study (1994–2018).

Characteristics	All Cases			Cases with Pre-Diagnostic Hypertension Systolic BP ≥ 130 mmHg		
	*n*	Hazard Ratio (95% C.I.)	*p*	*n*	Hazard Ratio (95% C.I.)	*p*
Age per 10 Years	142	1.13 (0.69–1.82)	0.63	87	1.17 (0.58–2.36)	0.66
*Capra-S*						
Low (0–2)	58	1 (Reference)		33	1 (Reference)	
Intermediate (3–5)	66	3.75 (1.17–8.27)	0.001	41	6.25 (1.41–27.3)	0.015
High (6–12)	21	16.0 (6.59–39.2)	<0.001	13	31.9 (6.50–156.5)	<0.001
*miR-24-1-5p*						
TE+NE low	82	1 (Reference)		55	1 (Reference)	
TE+NE high	63	1.99 (1.13–3.51)	0.017	32	2.85 (1.25–6.47)	0.013

* Adjusted for age, Capra-S group, MiR-24 expression, BMI, kg/m^2^, cholesterol, and alcohol use. Regarding miR-24-1-5p: low score < 1.57 and high score ≥ 1.57. Abbreviations: Sys, systolic; BP, blood pressure; CAPRA-S, Cancer of the Prostate Risk Assessment Postsurgical Score; CI, confidence interval; TE, tumor epithelium; NE, normal epithelium.

## Data Availability

Dataset available pending permission from the Tromsø Study. Please send request to first author.

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
