# Peer review of "Expression of miR-24-1-5p in Tumor Tissue Influences Prostate Cancer Recurrence: The PROCA-life Study"

_cancers, 2022, doi:10.3390/cancers14051142_

Round 1
Reviewer 1 Report
The title is interesting and it will attract readers from various fields such as cancer therapy and those who are working on molecular pathways and miRNAs. I suggest publication of current work. However, some issues should be addressed before publication. The first paragraph of introduction is about prostate cancer. However, it is too general and adds nothing to field. Be more specific and look at new articles to improve it. The abbreviation of microRNA is miRNA. Change it in all sections of manuscript. When you use miRNA with a number for instance miR-24-1-5p, it is ok to use miR. The second paragraph of introduction is about miRNAs. However, it is not informative. How miRNAs bind to mRNA? by binding to 3/-UTR? Why you have mentioned immune system? Not relevant to subject. About cancer, you have only mentioned aberrant expression occurs. This is no enough. Look at new article to improve this section (Doi, 10.1016/j.canlet.2021.03.025). Some of the statements do not have citation. For instance, last sentence of first paragraph in introduction. Is it possible to add a schematic figure in improving quality of current work? Only 3 or 4 articles from 2020. Please update references to improve quality and visibility of the work.
Reviewer 2 Report
The idea of the study is very interesting and promising. This study may demonstrate the potential of miR24-1-5p expression for separating prostate ca reoccurrence. However the study lacks of mechanism and there is no validation on normal prostate tissue miR24 expression. The authors only included examples for their staining and tis only comes from ISH data.
The authors need to provide more in depth analysis. Interpretation is weak and the following should be considered/added:
No further experiments were performed, including pathway analysis. No comments on higher/lower GS miR24 expression.
No info/discussion on perineural invasion and miR24 correlation
No info/discussion on BMI and miR24 expression
No info/discussion on all the other criteria that was included on table 1, patient demographic.
Any relation to the PSA values?
Any mechanistic suggestions to miR24-1-5p and inflammation?
Minor spellcheck and consistency is needed, for instance, it should be n not N (page 3, section 3.2).
Round 2
Reviewer 1 Report
The authors have appropriately responded to my comments. The only remained issue is that please remove microRNA from title and only miR-24-1-5p is enough.
Author Response
The title is updated as recommended.
Reviewer 2 Report
The ms was improved greatly, the authors added more info and evaluation. I would like to further suggest that since there is no correlation was detected between clinical parameters and miR 24 expression levels, authors should include the following studies to their discussion.
doi: 10.1038/s41389-017-0007-5.
doi: 10.3390/biology9030052.
doi: 10.21873/cgp.20108
Author Response
Thank you very much for your valuable feedback, and for giving us the opportunity to send you a revised version of the manuscript entitled. “Expression of microRNA miR-24-1-5p in tumor tissue influence prostate cancer recurrence. The PROCA-life Study.
We have read the useful comments and the articles suggested by the reviewer, and have carefully responded accordingly, and made the appropriate changes in the manuscript (marked with "track changes" in the text).
We have added sentences in the introduction and the discussion part, reflecting upon the content of the three articles (see introduction page 2 and discussion page 12 and page 13).
Looking forward to hear from you
Round 3
Reviewer 2 Report
Authors addressed all the comments/points.
Author Response
No further comments